# End-to-end Multi-modal Video Temporal Grounding

**Yi-Wen Chen**[1]     **Yi-Hsuan Tsai**[2]     **Ming-Hsuan Yang**[1,3,4]

[1]University of California, Merced     [2]Phiar     [3]Yonsei University     [4]Google Research

## Abstract

We address the problem of text-guided video temporal grounding, which aims to identify the time interval of a certain event based on a natural language description. Different from most existing methods that only consider RGB images as visual features, we propose a multi-modal framework to extract complementary information from videos. Specifically, we adopt RGB images for appearance, optical flow for motion, and depth maps for image structure. While RGB images provide abundant visual cues of certain events, the performance may be affected by background clutters. Therefore, we use optical flow to focus on large motion and depth maps to infer the scene configuration when the action is related to objects recognizable with their shapes. To integrate the three modalities more effectively and enable inter-modal learning, we design a dynamic fusion scheme with transformers to model the interactions between modalities. Furthermore, we apply intra-modal self-supervised learning to enhance feature representations across videos for each modality, which also facilitates multi-modal learning. We conduct extensive experiments on the Charades-STA and ActivityNet Captions datasets, and show that the proposed method performs favorably against state-of-the-art approaches.

## 1   Introduction

With the rapid growth of video data in our daily lives, video understanding has become an ever increasingly important task in computer vision. Research involving other modalities such as text and speech has also drawn much attention in recent years, *e.g.,* video captioning [17, 23], and video question answering [18, 16]. In this paper, we focus on text-guided video temporal grounding, which aims to localize the starting and ending time of a segment corresponding to a text query. It is one of the most effective approaches to understand video contents, and applicable to numerous tasks, such as video retrieval, video editing and human-computer interaction. This problem is considerably challenging as it requires accurate recognition of objects, scenes and actions, as well as joint comprehension of video and language.

Existing methods [34, 26, 33, 22] usually consider only RGB images as visual cues, which are less effective for recognizing objects and actions in videos with complex backgrounds. To understand the video contents more holistically, we propose a multi-modal framework to learn complementary visual features from RGB images, optical flow and depth maps. RGB images provide abundant visual information, which is essential for visual recognition. However, existing methods based on appearance alone are likely to be less effective for complex scenes with cluttered backgrounds. For example, since the query text descriptions usually involve moving objects such as "Closing a door" or "Throwing a pillow", using optical flow as input is able to identify such actions with large motion. On the other hand, depth is another cue that is invariant to color and lighting, and is often used to complement the RGB input in object detection and semantic segmentation. In our task, depth information helps the proposed model recognize actions involving objects with distinct shapes as the context. For example, actions such as "Sitting in a bed" or "Working at a table" are not easily recognized by optical flow due to small motion, but depth can provide structural information to assist the learning process. We also note that, our goal is to design an end-to-end multi-modal framework

35th Conference on Neural Information Processing Systems (NeurIPS 2021).

for video grounding by directly utilizing low-level cues such as optical flow and depth, while other alternatives based on object detector or semantic segmentation is out of the scope of this work.

To leverage multi-modal cues, one straightforward way is to construct a multi-stream model that takes individual modality as the input in each stream, and then averages the multi-stream output predictions to obtain final results. However, we find that this scheme is less effective due to the lack of communication across different modalities, *e.g.,* using depth cues alone without considering RGB features is not sufficient to learn the semantic information as the appearance cue does. To tackle this issue, we propose a multi-modal framework with 1) an inter-modal module that learns cross-modal features, and 2) an intra-modal module to self-learn feature representations across videos.

For inter-modal learning, we design a fusion scheme with co-attentional transformers [20] to dynamically fuse features from different modalities. One motivation is that, different videos may require to adopt a different combination of modalities, *e.g.,* "Working at a table" would require more appearance and depth information, while optical flow is more important for "Throwing a pillow". To enhance feature representations for each modality and thereby improve multi-modal learning, we introduce an intra-modal module via self-supervised contrastive learning [7, 15]. The goal is to ensure the feature consistency across video clips when they contain the same action. For example, with the same action "Eating", it may happen at different locations with completely different backgrounds and contexts, or with different text descriptions that "eats" different food. With our intra-modal learning, it enforces features close to each other when they describe the same action and learn features that are invariant to other distracted factors across videos, and thus it can improve our multi-modal learning paradigm.

We conduct extensive experiments on the Charades-STA [10] and ActivityNet Captions [17] datasets to demonstrate the effectiveness of our multi-modal learning framework for video temporal grounding using (D)epth, (R)GB, and optical (F)low with the (T)ext as the query, and name our method as *DRFT*. First, we present the complementary property of multi-modality and the improved performance over the single-modality models. Second, we validate the individual contributions of our proposed components, *i.e.,* inter- and intra-modal modules, that facilitate multi-modal learning. Finally, we show state-of-the-art performance for video temporal grounding against existing methods.

The main contributions of this work are summarized as follows: 1) We propose a multi-modal framework for text-guided video temporal grounding by extracting complementary information from RGB, optical flow and depth features. 2) We design a dynamic fusion mechanism across modalities via co-attentional transformer to effectively learn inter-modal features. 3) We apply self-supervised contrastive learning across videos for each modality to enhance intra-modal feature representations that are invariant to distracted factors with respect to actions.

## 2   Related Work

**Text-Guided Video Temporal Grounding.**   Given a video and a natural language query, text-guided video temporal grounding aims to predict the starting and ending time of the video clip that best matches the query sentence. Existing methods for this task can be categorized into two groups, *i.e.,* two-stage and one-stage schemes (see Figure 1(a)(b)). Most two-stage approaches adopt a propose-and-rank pipeline, where they first generate clip proposals and then rank the proposals based on their similarities with the query sentence. Early two-stage methods [10, 14] obtain proposals by scanning the whole video with sliding windows. Since the sliding window mechanism is computationally expensive and usually produces many redundant proposals, numerous methods are subsequently proposed to improve the efficiency and effectiveness of proposal generation. The TGN model [2] performs frame-by-word interactions and localize the proposals in one single pass. Other approaches focus on reducing redundant proposals by generating query-guided proposals [31] or semantic activity proposals [3]. The MAN method [34] models the temporal relationships between proposals using a graph architecture to improve the quality of proposals. To alleviate the computation of observing the whole video, reinforcement learning [13, 30] is utilized to guide the intelligent agent to glance over the video in a discontinuous way. While the two-stage methods achieve promising results, the computational cost is high for comparing all proposal-query pairs, and the performance is largely limited by the quality of proposal generation.

To overcome the issues of two-stage methods, some recent approaches adopt a one-stage pipeline to directly predict the temporal segment from the fusion of video and text features. Most of the one-stage approaches focus on the attention mechanisms or interaction between modalities. For

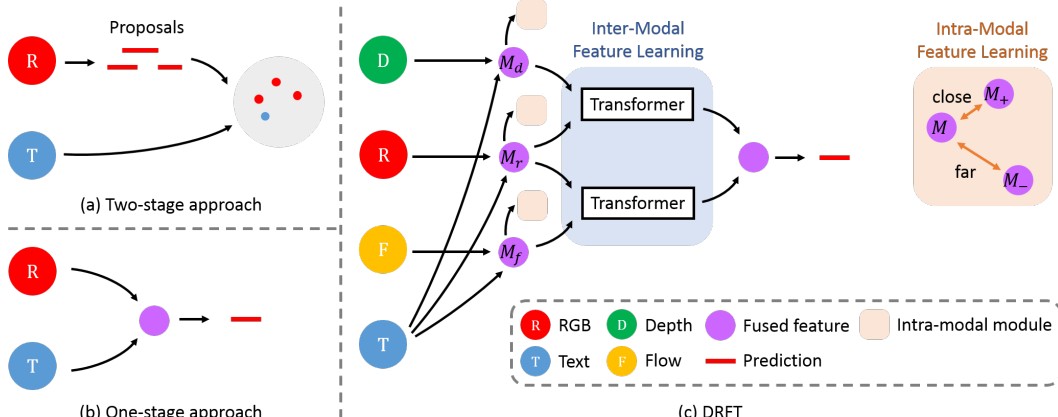

Figure 1: **Overview of the proposed algorithm.** Previous methods focus on (a) two-stage schemes relying on proposals, which is computational expensive and sensitive to the quality of proposals; (b) one-stage schemes predicting the temporal segment from the fusion of RGB and text features. In contrast, our DRFT method adopts the end-to-end one-stage scheme with multi-modal learning using RGB, depth, and optical flow, such that the model can better capture action-related features via our proposed inter-modal and intra-modal learning modules. Specifically, the inter-modal module uses co-attentional transformer to extract features across RGB and the other modality (either depth or optical flow), while the intra-modal module aims to perform cross-video feature learning, in which features within each modal are closer when they share the same action category.

example, the ABLR method [32] predicts the temporal coordinates using a co-attention based location regression algorithm. The ExCL mechanism [11] exploits the cross-modal interactions between video and text, and the PfTML-GA model [26] improves the performance by introducing the query-guided dynamic filter. Moreover, the DRN scheme [33] leverages dense supervision from the sparse annotations to facilitate the training process. Recently, the LGI model [22] decomposes the query sentence into multiple semantic phrases and conducts local and global interactions between the video and text features. In our framework, we adopt LGI as the baseline that uses the hierarchical video-text interaction. However, different from LGI that only considers RGB frames as input, we take RGB, optical flow and depth as input, and design the inter-modality learning technique to learn complementary information from the video. Furthermore, we apply contrastive learning across videos to enhance the feature representations in each modality, which helps the learning of the whole model (see Figure 1(c)).

**Multi-Modal Learning.** As typical event or actions can be described by signals from multiple modalities, understanding the correlation between different modalities is crucial to solve problems more comprehensively. Research on joint vision and language learning [6, 23, 16, 7] has gained much attention in recent years since natural language is an intuitive way for human communication. Recent studies [24, 8] based on the transformer [29] have shown great success in self-supervised learning and transfer learning for natural language tasks. The transformer-based BERT model [8] has also been widely used to learn joint representations for vision and language. These methods [20, 21, 35, 19, 5, 9] aim to learn generic representations from a large amount of image-text pairs in a self-supervised manner, and then fine-tune the model for downstream vision and language tasks. The ViLBERT scheme [20] extracts features from image and text using two parallel BERT-style models, and then connects the two streams with the co-attentional transformer layers. In this work, we focus on the video temporal grounding task guided by texts, while introducing multi-modality to improve model learning, which is not studied before. For fusing the multi-modal information, we leverage the co-attentional transformer layers [20] in our framework and design an approach by fusing the RGB features with optical flow and depth features respectively.

## 3 Proposed Framework

In this work, we address the problem of text-guided video temporal grounding using a multi-modal framework. The pipeline of the proposed framework is illustrated in Figure 2. Given an input video $V = \{V_t\}_{t=1}^T$ with $T$ frames and a query sentence $Q = \{Q_i\}_{i=1}^N$ with $N$ words, we aim to localize

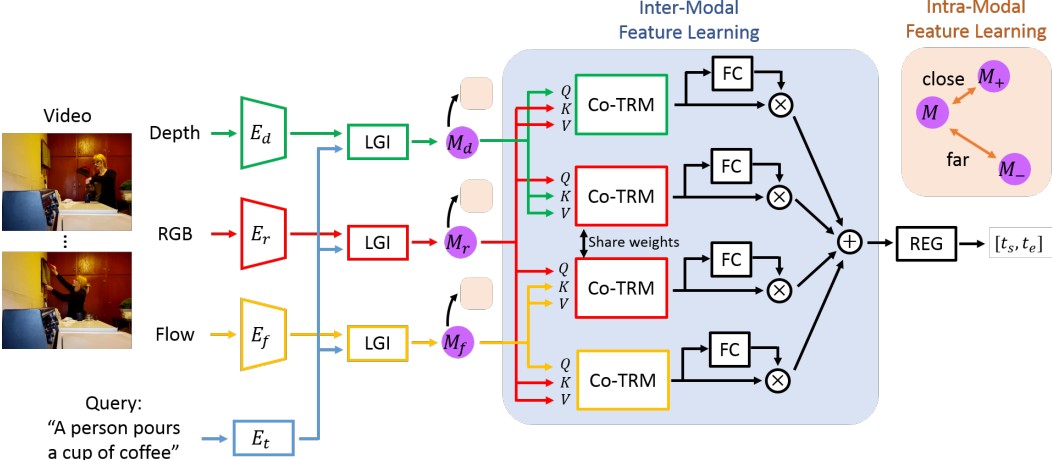

Figure 2: **Pipeline of the proposed DRFT framework.** Given an input video, our goal is to predict the video segment $[t_s, t_e]$ that contains the action in the guided query text. To this end, we first feed multi-modal information, *i.e.,* depth, RGB, and optical flow to individual encoders, $E_d$, $E_r$, $E_f$, and extract their features. Then we use the local-global interaction modules (LGI) to attend the text features via the text encoder $E_t$ on individual modalities, and obtain the multi-modal features $M_d$, $M_r$ and $M_f$ for depth, RGB and flow respectively. To fuse multi-modal features effectively, we propose an inter-modal module that uses the co-attentional transformers (Co-TRM) to learn features between RGB and the other modality (either depth or optical flow), and then dynamically fuses the multi-modal features, *i.e.,* the $\otimes$ and $\oplus$ operations with FC layers to predict the weights. To enhance feature representations within each module, we incorporate an intra-modal module that enforces cross-video features in each modality to be close to each other when they contain the same action, otherwise they should be apart from each other.

the starting and ending time $[t_s, t_e]$ of the event corresponding to the query. To this end, we design a multi-modal framework to learn complementary visual information from RGB images, optical flow and depth maps. From the input video, we first compute the depth map of each frame and the optical flow of each pair of consecutive frames. We then apply the visual encoders $E_d$, $E_r$, $E_f$ to extract features from the depth, RGB and flow inputs. A textual encoder $E_t$ is utilized to extract the feature of the query sentence $Q$. The local-global interaction modules (LGI) then incorporate the textual feature into each visual modality, and generate the multi-modal features $M_d$, $M_r$ and $M_f$ for depth, RGB and flow respectively.

To effectively integrate the features from different modalities and enable inter-modal feature learning, we propose a dynamic fusion scheme with transformers to model the interaction between modalities. The feature after integration is then fed into a regression module (REG) to predict the starting and ending time $[t_s, t_e]$ of the target video segment. To enhance the feature representations in each modality, we introduce an intra-modal learning module that conducts self-supervised contrastive learning across videos. The intra-modal learning is applied on the multi-modal features $M_d$, $M_r$ and $M_f$ separately to enforce features of video segments containing the same action to be close to each other, and those from different action categories to be far apart.

### 3.1 Inter-Modal Feature Learning

Videos contain rich information in both spatial and temporal dimensions. To learn information more comprehensively, in addition to the RGB modality, we also consider optical flow that captures motion, and depth feature that represents image structure. An intuitive way to combine the three modalities is to utilize a multi-stream model and directly average the outputs of individual streams. However, since the importance of each modality is not the same in different situations, directly averaging them may downweigh the importance of a specific modality and degrade the performance. In Table 1, we present the results of two-stream (RGB and flow) and three-stream (RGB, flow and depth) baseline models, where the outputs from different modalities are averaged before the final output layer. Compared to the single-stream (RGB) baseline model, the multi-stream models do not improve the performance, which shows that it is not intuitive to learn complementary information from multi-modal features.

Table 1: **Performance of baseline methods.** For the single modality, we use RGB only, while using RGB and flow as the two modalities. More baselines results are provided in the supplementary material.

| Method | Charades-STA | | | | ActivityNet Captions | | | |
| --- | --- | --- | --- | --- | --- | --- | --- | --- |
| | R@0.3 | R@0.5 | R@0.7 | mIoU | R@0.3 | R@0.5 | R@0.7 | mIoU |
| Single-stream baseline | 72.14 | 59.16 | 35.31 | 50.94 | 58.47 | 41.35 | 23.60 | 41.38 |
| Two-stream baseline | 71.57 | 58.35 | 35.72 | 50.34 | 57.78 | 40.51 | 22.66 | 40.70 |
| Three-stream baseline | 71.13 | 57.39 | 34.69 | 48.21 | 56.45 | 38.63 | 22.05 | 39.86 |

Such cases may happen frequently in certain actions. For example, flow features would not help much for "Sitting in a bed" but would help more for "Closing a door". Therefore, having a dynamic mechanism is critical for multi-modal fusion.

**Co-attentional Feature Fusion.** The ensuing question becomes how to learn effective features across modalities and also fuse them dynamically. First, we observe that, although depth and flow modalities are effective in some situations, they alone are not able to capture the semantic information, which is crucial for video-text understanding. Thereby, we design a co-attentional scheme to allow joint feature learning between RGB and another modality (either depth or flow).

Inspired by the co-attentional transformer layer [20] that consist of multi-headed attention blocks, where it takes a paired feature as the input (*e.g.,* $M_d$ and $M_r$) and forms three matrices, Q, K, and V that represent queries, keys, and values (also see Figure 2). In this way, the multi-headed attention block for each modality takes the keys and values from the other modality as the input, and thus outputs the attention-pooled features conditioned on the other modality. For instance, we consider the pair of $M_d = \{m_d^1, ..., m_d^T\}$ and $M_r = \{m_r^1, ..., m_r^T\}$ features for depth and RGB, where $T$ is the number of frames in a video clip. The co-attentional transformer layer performs depth-conditioned RGB feature attention, as well as RGB-conditioned depth feature attention. Similarly, for flow and RGB features, $M_f = \{m_f^1, ..., m_f^T\}$ and $M_r = \{m_r^1, ..., m_r^T\}$, we obtain another set of flow-conditioned RGB feature attention and RGB-conditioned flow feature attention. Note that, since RGB feature generally contains the most abundant information in the video and is used for both depth and flow attention, we adopt a shared multi-headed attention block for RGB as shown in Figure 2.

**Dynamic Feature Fusion.** To effectively combine these output features from co-attentional transformers and perform the final prediction, we dynamically learn the weights for each multi-modal feature and linearly combine the four features using the weights (see Figure 2). Since the importance of each modality depends on the input data, we generate the weights by feeding each feature into a fully-connected (FC) layer, and normalize the weights to make the sum equal to 1. By dynamically generating weights from the features, we are able to adapt the multi-modal fusion process according to the input video and the query text. The fused feature is then served as input of the regression module (REG) to predict the starting and ending time.

### 3.2 Intra-Modal Feature Learning

To facilitate the multi-modal training, we introduce an intra-modal feature learning module, which enhances feature representations within each modality by applying self-supervised contrastive learning. Our motivation is that features in the same action category should be similar even if they are from different videos. To this end, for each input video $V$, we randomly sample positive videos $V_+$ that contain the same action category, and negative videos $V_-$ with different action categories. We perform contrastive learning on the multi-modal features $M_d$, $M_r$ and $M_f \in \mathbb{R}^{c \times T}$ separately for each modality, where $c$ is the feature dimension and $T$ is the number of frames. Since the multi-modal feature contains information from the whole video, we only consider features that contain the action by extracting the corresponding video segment. We then conduct average pooling in the temporal dimension and obtain a feature vector $M \in \mathbb{R}^c$. The contrastive loss $L_{cl}$ is formulated as:

$$L_{cl} = -\log \frac{\sum\limits_{M_+ \in Q_+} e^{h(M)^\top h(M_+)/\tau}}{\sum\limits_{M_+ \in Q_+} e^{h(M)^\top h(M_+)/\tau} + \sum\limits_{M_- \in Q_-} e^{h(M)^\top h(M_-)/\tau}}, \tag{1}$$

where $Q_+$ and $Q_-$ are the sets of positive and negative samples, and $\tau$ is the temperature parameter. Following the SimCLR approach [4], we use a linear layer $h(\cdot)$ to project the feature $M$ to another embedding space where we apply contrastive loss. We accumulate the loss from each modality to be the final contrastive loss, namely $L_{cl} = L_{cl}^r(h_r(AvgPool(M_r))) + L_{cl}^d(h_d(AvgPool(M_d))) + L_{cl}^f(h_f(AvgPool(M_f)))$ for RGB, depth, and flow, respectively.

### 3.3 Model Training and Implementation Details

**Overall Objective.** The overall objective of the proposed method is composed of the supervised loss $L_{grn}$ for predicting temporal grounding that localizes the video segment and the self-supervised contrastive loss $L_{cl}$ for intra-modal learning in (1): $L = L_{grn} + L_{cl}$. The supervised loss $L_{grn}$ is the same as the loss defined in the LGI method [22], which includes:

1) Location regression loss $L_{reg} = smooth_{L1}(\hat{t}^s - t^s) + smooth_{L1}(\hat{t}^e - t^e)$ that calculates the L1 distance between the normalized ground truth time interval $(\hat{t}^s, \hat{t}^e) \in [0, 1]$ and the predicted time interval $(t^s, t^e)$, where $smooth_{L1}$ is defined as $0.5x^2$ if $|x| < 1$ and $|x| - 0.5$ otherwise.

2) Temporal attention guidance loss $L_{tag} = -\dfrac{\sum\limits_{i=1}^{T} \hat{\mathbf{o}}_i \log(\mathbf{o}_i)}{\sum\limits_{i=1}^{T} \hat{\mathbf{o}}_i}$ for the temporal attention in the REG module, where $\hat{\mathbf{o}}_i$ is set to 1 if the $i$-th segment is located within the ground truth time interval and 0 otherwise.

3) Distinct query attention loss $L_{dqa} = ||(\mathbf{A}^\top \mathbf{A}) - \lambda I||_F^2$ to enforce query attention weights to be distinct along different steps in the LGI module, where $\mathbf{A} \in \mathbb{R}^{N \times S}$ is the concatenated query attention weights across $S$ steps, $|| \cdot ||_F$ denotes Frobenius norm of a matrix, and $\lambda \in [0, 1]$ controls the extent of overlap between query attention distributions. The supervised loss is the sum of the three loss terms $L_{grn} = L_{reg} + L_{tag} + L_{dqa}$ and we use the default setting in LGI [22], in which we refer readers to their paper for more details.

**Implementation Details.** We generate optical flow and depth maps using the RAFT [27] and MiDaS [25] method respectively. For the visual encoder $E_d$, $E_r$ and $E_f$, we employ the I3D [1] and C3D [28] networks for Charades-STA and ActivityNet Captions datasets respectively. As for the textual encoder $E_t$, we adopt a bi-directional LSTM, where the feature is obtained by concatenating the last hidden states in forward and backward directions. The LGI module in our framework contains the sequential query attention and local-global video-text interactions as in the LGI model [22]. The REG module generates temporal attention weights to aggregate the features and performs regression via an MLP layer. The operations are defined similar to LGI [22]. The feature dimension $c$ is set to 512. In the contrastive loss (1), the temperature parameter $\tau$ is set to 0.1. The projection head $h(\cdot)$ is a 2-layer MLP that project the feature to a 512-dimensional latent space. We implement the proposed model in PyTorch with the Adam optimizer and a fixed learning rate of $4 \times 10^{-4}$. The source code and models are available at https://github.com/wenz116/DRFT.

## 4 Experimental Results

### 4.1 Datasets and Evaluation Metric

We evaluate the proposed DRFT method against the state-of-the-art approaches on two benchmark datasets, *i.e.,* Charades-STA [10] and ActivityNet Captions [17].

**Charades-STA.** It is built upon the Charades dataset for evaluating the video temporal grounding task. It contains 6,672 videos involving 16,128 video-query pairs, where 12,408 pairs are used for training and 3,720 pairs are for testing. The average length of the videos is 29.76 seconds. There are 2.4 annotated moments with duration 8.2 seconds in each video.

**ActivityNet Captions.** It is originally constructed for dense video captioning from the ActivityNet dataset. The captions are used as queries in the video temporal grounding task. It consists of 20k YouTube videos with an average duration of 120 seconds. The videos are annotated with 200 activity categories, which is more diverse compared to the Charades-STA dataset. Each video contains 3.65 queries, where each query has an average length of 13.48 words. The dataset is split into training, validation and testing set with a ratio of 2:1:1, resulting in 37,421, 17,505 and 17,031 video-query

Table 2: **Comparisons with state-of-the-art methods.** For the results on ActivityNet Captions, DRN [33] is evaluated on the two validation sets $val_1$ and $val_2$ separately, while others are evaluated on the entire validation set. We also note that the single-modal model is RGB only, while RGB and flow are used in the two-modality case. More results are presented in the supplementary material.

| Method | Charades-STA | | | | ActivityNet Captions | | | |
| | R@0.3 | R@0.5 | R@0.7 | mIoU | R@0.3 | R@0.5 | R@0.7 | mIoU |
|---|---|---|---|---|---|---|---|---|
| CTRL [10] | - | 21.42 | 7.15 | - | 28.70 | 14.00 | 20.54 | - |
| RWM [13] | - | 36.70 | - | - | - | 36.90 | - | - |
| MAN [34] | - | 46.53 | 22.72 | - | - | - | - | - |
| TripNet [12] | 51.33 | 36.61 | 14.50 | - | 45.42 | 32.19 | 13.93 | - |
| PfTML-GA [26] | 67.53 | 52.02 | 33.74 | - | 51.28 | 33.04 | 19.26 | 37.78 |
| DRN [33] | - | 53.09 | 31.75 | - | - | 42.49/45.45 | 22.25/24.36 | - |
| LGI [22] | 72.96 | 59.46 | 35.48 | 51.38 | 58.52 | 41.51 | 23.07 | 41.13 |
| Single-stream DRFT | 73.85 | 60.79 | 36.72 | 52.64 | 60.25 | 42.37 | 25.23 | 43.18 |
| Two-stream DRFT | 74.26 | 61.93 | 38.69 | 53.92 | 61.80 | 43.71 | 26.43 | 44.82 |
| Three-stream DRFT | **76.68** | **63.03** | **40.15** | **54.89** | **62.91** | **45.72** | **27.79** | **45.86** |

pairs respectively. Since the testing set is not publicly available, we follow previous methods to evaluate the performance on the combination of the two validation sets, which are denoted as $val_1$ and $val_2$.

Following the typical evaluation setups [10, 22], we employ two metrics to assess the performance of video temporal grounding: 1) Recall at various thresholds of temporal Intersection over Union (R@IoU). It measures the percentage of predictions that have IoU with the ground truth larger than the threshold. We adopt 3 values {0.3, 0.5, 0.7} for the IoU threshold. 2) mean tIoU (mIoU). It is the average IoU over all results.

## 4.2   Overall Performance

In Table 2, we evaluate our framework against state-of-art approaches, including two-stage methods that rely on propose-and-rank schemes [10, 13, 34] and one-stage methods that only consider RGB videos as the input [12, 26, 33, 22]. First, compared to our baseline LGI [22], our results with single/two/three modalities are consistently better than theirs in all the evaluation metrics, which demonstrates the benefit of our intra-modal feature learning scheme and the inter-modal feature fusion mechanism. We also note that for our single-modal model, we use RGB as the input and only apply the intra-modal contrastive learning across videos, where it already performs favorably against existing algorithms. More results of the single-stream models using other modalities are provided in the supplementary material.

Second, we show that with the increased modality used in our model (bottom group in Table 2), the performance on two benchmarks are consistently improved, which demonstrates the complementary property of RGB, depth, and flow for video temporal grounding. Moreover, compared to the worse baseline results when adding more modalities without our proposed modules in Table 1, we validate the importance of designing a proper scheme of exchanging and fusing the information across modalities. It is also worth mentioning that with more modalities involved in the model, our method achieves larger performance gains compared to the baseline, *e.g.,* more than 5% improvement in all the metrics on two benchmarks.

## 4.3   Ablation Study

In Table 3, we present the ablation of individual components proposed in our framework.

**Inter-modal Feature Fusion.**   To enhance the communication across modalities, we propose to first use co-attentional transformers to learn attentive features across RGB and another modality, and then use a dynamic feature fusing scheme with learned weights to combine different features. In the first four rows of the middle group in Table 3, we show the following properties in this work:

Table 3: **Ablation study of the proposed DRFT algorithm.** We present the importance of the proposed modules in our framework, namely 1) the inter-modal module that includes cross-modal feature learning via transformers and a dynamic feature fusion scheme via learnable weights; 2) the intra-modal feature learning method via cross-video contrastive feature learning for individual modalities.

| Method | Charades-STA | | | | ActivityNet Captions | | | |
|---|---|---|---|---|---|---|---|---|
| | R@0.3 | R@0.5 | R@0.7 | mIoU | R@0.3 | R@0.5 | R@0.7 | mIoU |
| Three-stream baseline | 71.13 | 57.39 | 34.69 | 48.21 | 56.45 | 38.63 | 22.05 | 39.86 |
| DRFT w/o transformer | 74.72 | 61.05 | 38.26 | 52.74 | 61.04 | 43.83 | 25.74 | 43.90 |
| DRFT w/ flow-RGB, flow-depth | 74.96 | 61.20 | 38.57 | 53.01 | 61.39 | 43.94 | 25.85 | 44.17 |
| DRFT w/o Co-TRM weight sharing | 76.24 | 62.61 | 39.60 | 54.47 | 62.59 | 45.28 | 27.36 | 45.31 |
| DRFT w/o learnable weights | 75.03 | 61.65 | 38.78 | 53.11 | 61.47 | 44.42 | 26.31 | 44.39 |
| DRFT w/o contrastive loss | 75.41 | 61.87 | 39.02 | 53.65 | 61.59 | 44.50 | 26.48 | 44.61 |
| Three-stream DRFT | 76.68 | 63.03 | 40.15 | 54.89 | 62.91 | 45.72 | 27.79 | 45.86 |

1) Using transformers is effective for multi-modal feature learning. As shown in the first row of the middle group, the performance drops without the co-attentional transformers for feature fusion.

2) RGB information is essential for the temporal grounding task, and thus we conduct co-attention between a) RGB-flow and b) RGB-depth. In the second row of the middle group, if using the flow modality as the common modality, *i.e.,* flow-RGB and flow-depth, the performance is worse than our final model.

3) Since RGB features are used for both flow and depth attention, we adopt a shared co-attention block for RGB as shown in Figure 2, where it can take RGB together with either the flow or depth cue as the input, and further enriches the attention mechanism. This design has not been considered in the prior work. In the third row of the middle group, without sharing the co-attentional module, the performance is worse than our final model.

4) The proposed dynamic fusion scheme via learnable weights is important to fuse features from different modalities. As shown the fourth row of the middle group, the performance drops significantly without learnable weights. Interestingly, learning dynamic weights to combine features is almost equally important compared to feature learning via transformers. This indicates that even with the state-of-the-art feature attention module, it is still challenging to combine multi-modal features.

**Intra-modal Feature Learning.** In the last row of the middle group in Table 3, we show the benefit of having the intra-modal cross-video feature learning. While multi-modal feature fusion already provides a strong baseline in our framework, improving feature representations in individual modalities is still critical for enhancing the entire multi-modal learning paradigm, in which such observations are not widely studied yet.

**Qualitative Results.** In Figure 3, we show sample results on the Charades-STA and ActivityNet Captions datasets, where the arrows indicate the starting and ending points of the corresponding grounded segment based on the query. Compared to the baseline method that only consider RGB features, the proposed DRFT approach is able to predict more accurate results by leveraging the multi-modal features from RGB, optical flow and depth. More results are presented in the supplementary material.

### 4.4 Analysis of Multi-modal Learning

To understand the complementary property of each modality, we analyze the video temporal grounding results of some example action categories. Figure 4 shows the performance of the single-stream baseline with RGB as input, single-stream DRFT models with RGB, flow or depth as input, and three-stream DRFT model respectively. The three plots contain categories where RGB, flow or depth performs better than the other two modalities. We first show that the single-stream DRFT model with contrastive learning improves from the single-stream baseline (red bars vs. orange bars). We then investigate the complementary property between the three modalities. For actions with smaller movement (*e.g.,* "smiling") in the left group of Figure 4, models using RGB as input generally

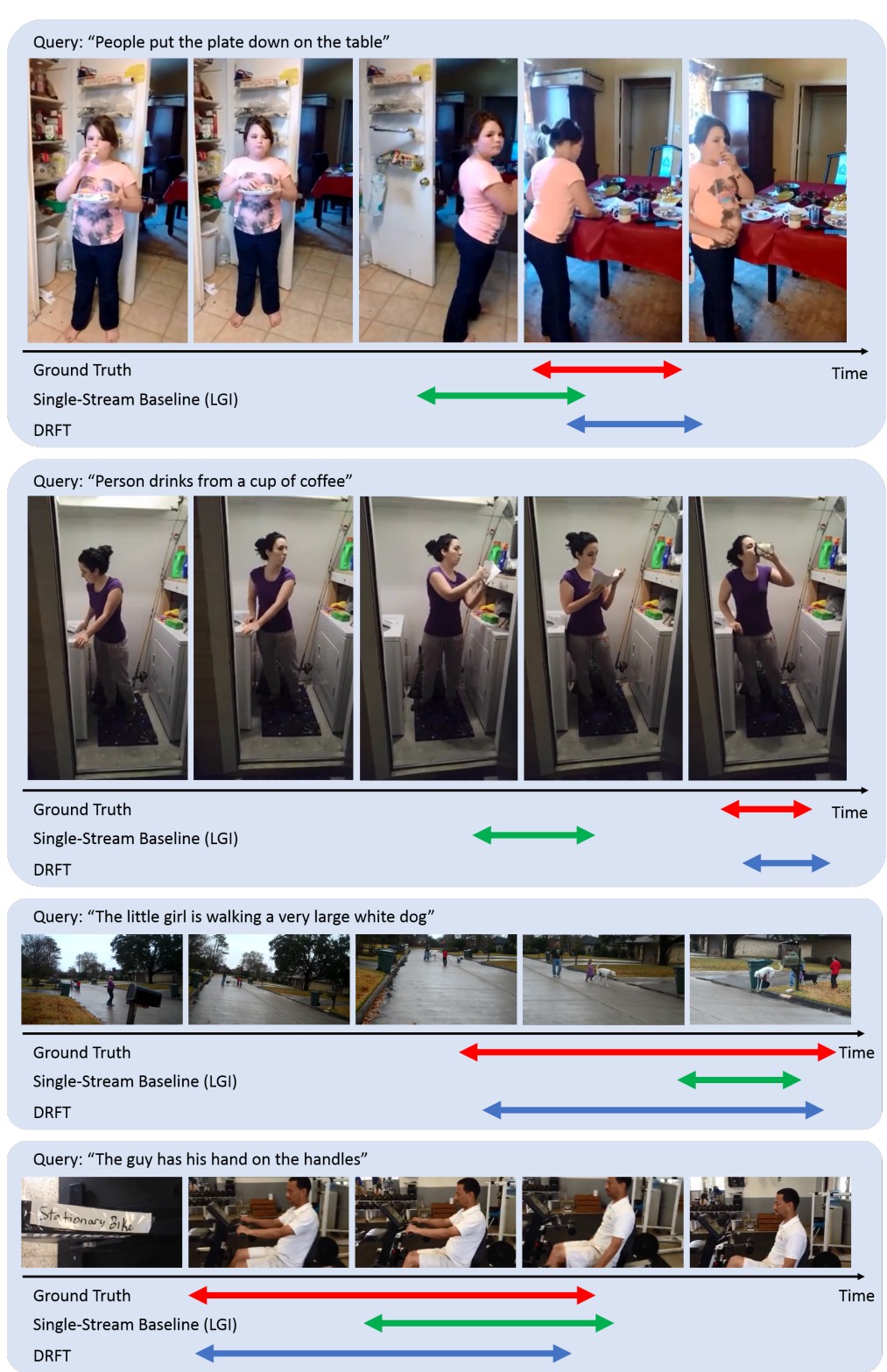

Figure 3: **Sample results on the Charades-STA (first and second rows) and ActivityNet Captions (third and fourth rows) datasets.** We show that the prediction of the proposed DRFT model is more accurate than the single-stream baseline (LGI [22]) with RGB as input. The arrows indicate the starting and ending points of the grounded video segment based on the query.

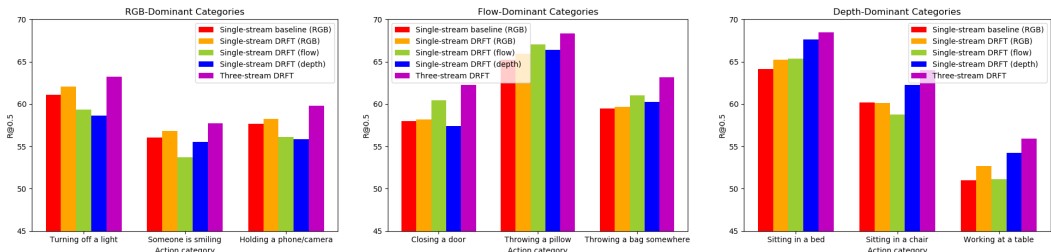

Figure 4: **Category-wise results on the Charades-STA dataset.** The three plots show categories where RGB, flow or depth perform better than the other two modalities.

Table 4: **Weights of dynamic fusion.** The → means attention from one to another modality.

| Action category | Flow → RGB | RGB → Flow | Depth → RGB | RGB → Depth |
|---|---|---|---|---|
| Turning off a light | **0.384** | 0.208 | 0.261 | 0.147 |
| Someone is smiling | **0.327** | 0.215 | 0.293 | 0.165 |
| Holding a phone/camera | **0.353** | 0.220 | 0.240 | 0.187 |
| Closing a door | 0.305 | **0.339** | 0.217 | 0.139 |
| Throwing a pillow | 0.312 | **0.379** | 0.194 | 0.115 |
| Throwing a bag somewhere | 0.280 | **0.362** | 0.203 | 0.155 |
| Sitting in a bed | 0.205 | 0.185 | 0.236 | **0.374** |
| Sitting in a chair | 0.221 | 0.173 | 0.284 | **0.322** |
| Working at a table | 0.216 | 0.158 | 0.262 | **0.364** |

perform better. For actions with larger motion (*e.g.,* "closing a door" or "throwing a pillow") in the middle group, flow provides more useful information (denoted as green bars). As for the actions with small motion but can be easily recognized by their structure (*e.g.,* "sitting in a bed" or "working at a table") in the right group, depth is superior to the other two modalities (denoted as blue bars). With the complementary property between RGB, flow and depth, we can take advantage of each modality and further improve the performance in the three-stream DRFT model (denoted as purple bars).

To further analyze the impact of each modality, we provide the learned weights for dynamic fusion in the three-stream DRFT for these categories in Table 4, where the top, middle and bottom groups contain categories that RGB, flow and depth help the most respectively. Flow → RGB means flow-conditioned RGB features, etc. We observe that for actions with smaller movement (top group), the weights for RGB features are larger. For actions with larger motion (middle group), the weights for optical flow are larger. Regarding actions with small motion but can be easily recognized by their structure (bottom group), the weights for depth are larger. This shows that the model can exploit each modality based on the complementary property between RGB, flow and depth.

## 5   Conclusions

In this paper, we focus on the task of text-guided video temporal grounding. In contrast to existing methods that consider only RGB images as visual features, we propose the DRFT model to learn complementary visual information from RGB, optical flow and depth modality. While RGB features provide abundant appearance information, we show that representation models based on these cues alone are not effective for temporal grounding in videos with cluttered backgrounds. We therefore adopt optical flow to capture motion cues, and depth maps to estimate image structure. To combine the three modalities more effectively, we propose an inter-modal feature learning module, which performs co-attention between modalities using transformers, and dynamically fuses the multi-modal features based on the input data. To further enhance the multi-modal training, we incorporate an intra-modal feature learning module that performs self-supervised contrastive learning within each modality. The contrastive loss enforces cross-video features to be close to each other when they contain the same action, and be far apart otherwise. We conduct extensive experiments on two benchmark datasets, demonstrating the effectiveness of the proposed multi-modal framework with inter- and intra-modal feature learning.

## Acknowledgements

This work is supported in part by NSF CAREER grant 1149783 and gifts from Snap as well as eBay.

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
