# End-to-end Multi-modal Video Temporal Grounding

**Yi-Wen Chen**[1]     **Yi-Hsuan Tsai**[2]     **Ming-Hsuan Yang**[1,3,4]
[1]University of California, Merced     [2]Phiar     [3]Yonsei University     [4]Google Research

In this supplementary document, we provide additional analysis and experimental results, including 1) more implementation details, 2) results of single-stream and two-stream models, and 3) more qualitative results for text-guided video temporal grounding.

## 1   Implementation Details

The co-attentional transformer layers in our model have hidden states of size 1024 and 8 attention heads. For the intra-modality contrastive learning, we randomly sample 3 positive videos that contain the same action as the anchor video and 4 negative videos with different action categories. Our framework is implemented on a machine with an Intel Xeon 2.3 GHz processor and an NVIDIA GTX 1080 Ti GPU with 11 GB memory.

## 2   Results of Single-stream and Two-stream Models

In Table 1, we present the results of single-stream models using optical flow or depth as visual input, and two-stream models using depth and RGB or depth and optical flow as the two modalities. Compared to the single-stream baseline using either flow or depth as input, our full models (DRFT) with contrastive loss consistently improve the performance, showing the effectiveness of the intra-modal contrastive learning mechanism. For the two-stream models where depth and RGB or depth and flow are the two input modalities, the performance gains from baseline to full model are larger when both inter- and intra-modal learning techniques are applied. We also observe that, compared to the other two modalities, depth is not the strongest one but is still complementary to other modalities. Moreover, the proposed DRFT method can always improve the baseline performance.

Table 1: **Performance of single-stream and two-stream methods.** The single-stream models use optical flow (F) or depth (D) as input, and the two-stream models take depth (D) and RGB (R) or depth (D) and optical flow (F) as the two input modalities.

| Method | Visual Feat. | Charades-STA | | | | ActivityNet Captions | | | |
|---|---|---|---|---|---|---|---|---|---|
| | | R@0.3 | R@0.5 | R@0.7 | mIoU | R@0.3 | R@0.5 | R@0.7 | mIoU |
| Single-stream baseline | F | 70.36 | 56.68 | 34.42 | 48.35 | 56.69 | 38.57 | 21.32 | 39.64 |
| Single-stream DRFT | F | 71.67 | 57.92 | 36.23 | 49.71 | 57.84 | 39.60 | 22.52 | 40.89 |
| Single-stream baseline | D | 70.17 | 55.93 | 34.20 | 47.68 | 56.52 | 39.23 | 21.26 | 38.47 |
| Single-stream DRFT | D | 71.54 | 57.78 | 35.92 | 49.55 | 57.59 | 40.64 | 22.61 | 40.12 |
| Two-stream baseline | D, R | 71.34 | 57.51 | 35.49 | 48.70 | 57.43 | 40.46 | 22.35 | 39.69 |
| Two-stream DRFT | D, R | 73.72 | 60.93 | 38.12 | 52.75 | 61.18 | 43.29 | 25.86 | 43.61 |
| Two-stream baseline | D, F | 70.94 | 57.21 | 35.08 | 48.56 | 57.13 | 39.82 | 21.79 | 39.56 |
| Two-stream DRFT | D, F | 72.83 | 59.32 | 37.26 | 50.81 | 59.34 | 42.08 | 23.96 | 41.84 |

## 3 Qualitative Results

We provide more qualitative results on the Charades-STA [1] and ActivityNet Captions [2] datasets in Figure 1, 2 and 3. Compared to the baseline LGI [3] that only takes RGB as input, the proposed DRFT model can generate more accurate results in difficult situations such as cluttered backgrounds (first two cases in Figure 1), dark environments (third case in Figure 2) or small motion (last case in Figure 3). We also show some failure cases of our method in Figure 4. In the first case, the actual span of the query action "putting some clothes onto some clothing racks" is longer than the annotation, where our result also covers most of the event. For the second video, the person walks through the doorway twice, but only the first one is annotated, while our model predicts the second one. In these cases, the baseline LGI model also fails to generate results that match the ground truth.

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

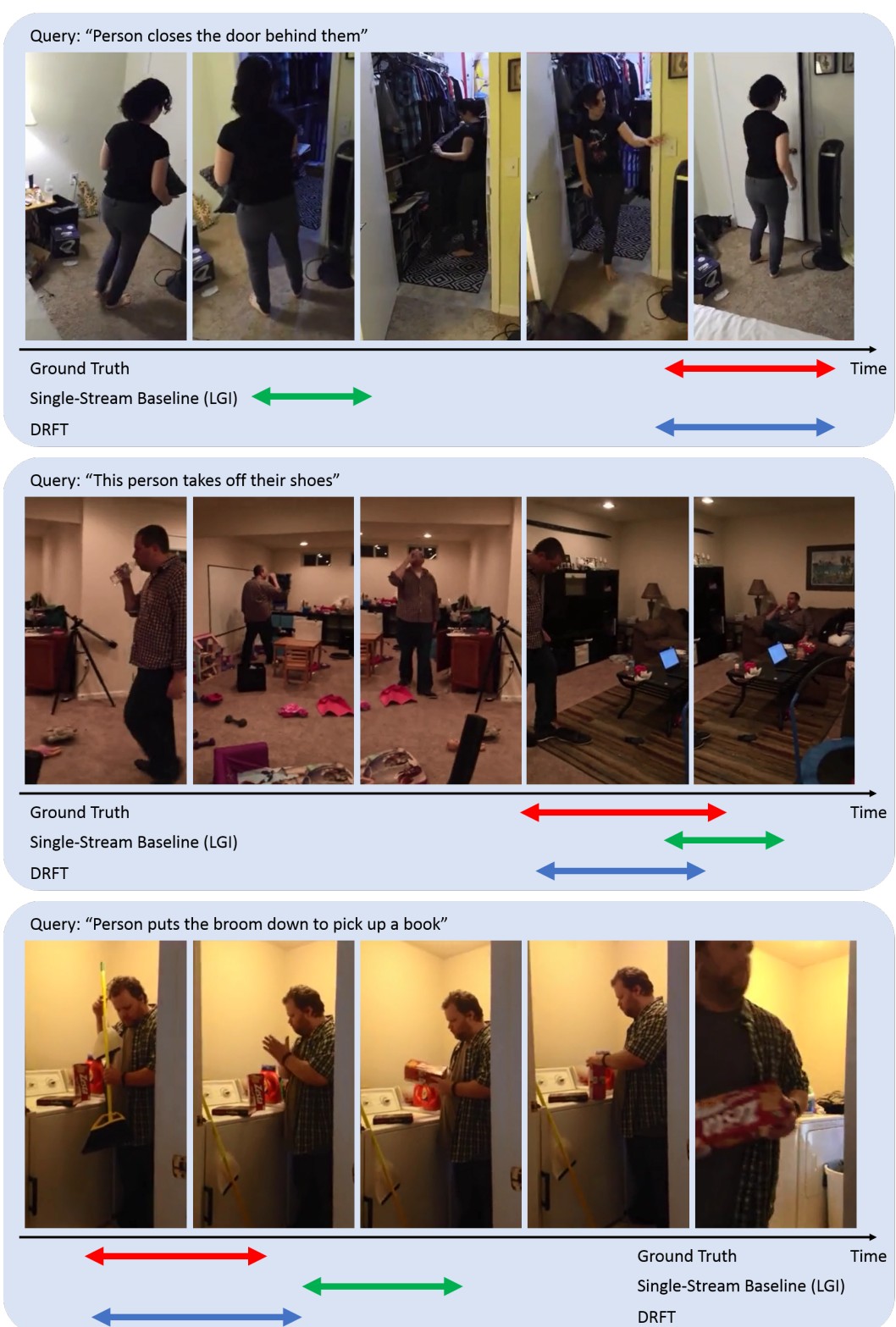

Figure 1: **Sample results on the Charades-STA dataset.** We show that the prediction of the proposed DRFT model is more accurate than the single-stream baseline (LGI [3]) with RGB as input. The arrows indicate the starting and ending points of the grounded video segment based on the query.

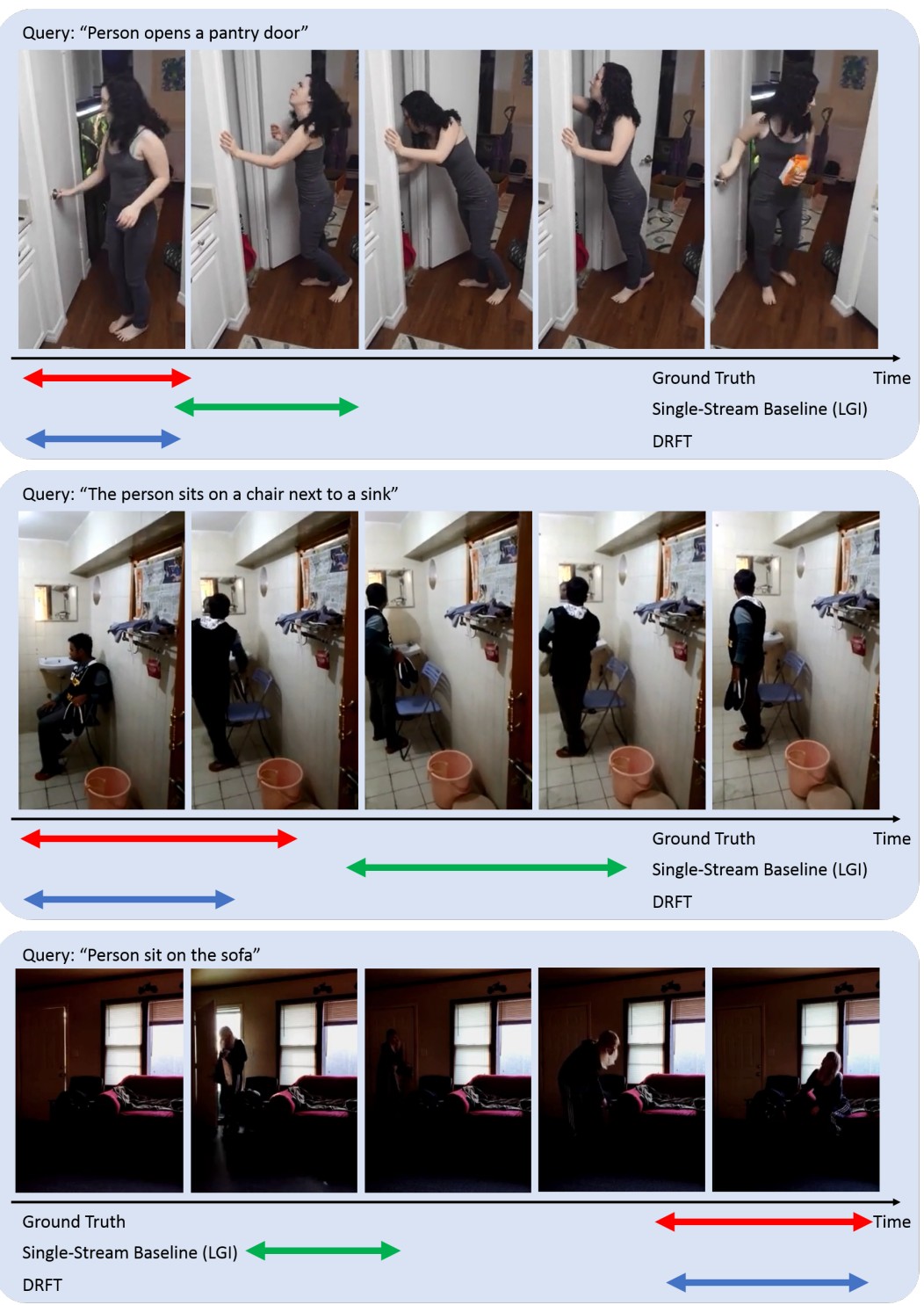

Figure 2: **Sample results on the Charades-STA dataset.** We show that the prediction of the proposed DRFT model is more accurate than the single-stream baseline (LGI [3]) with RGB as input. The arrows indicate the starting and ending points of the grounded video segment based on the query.

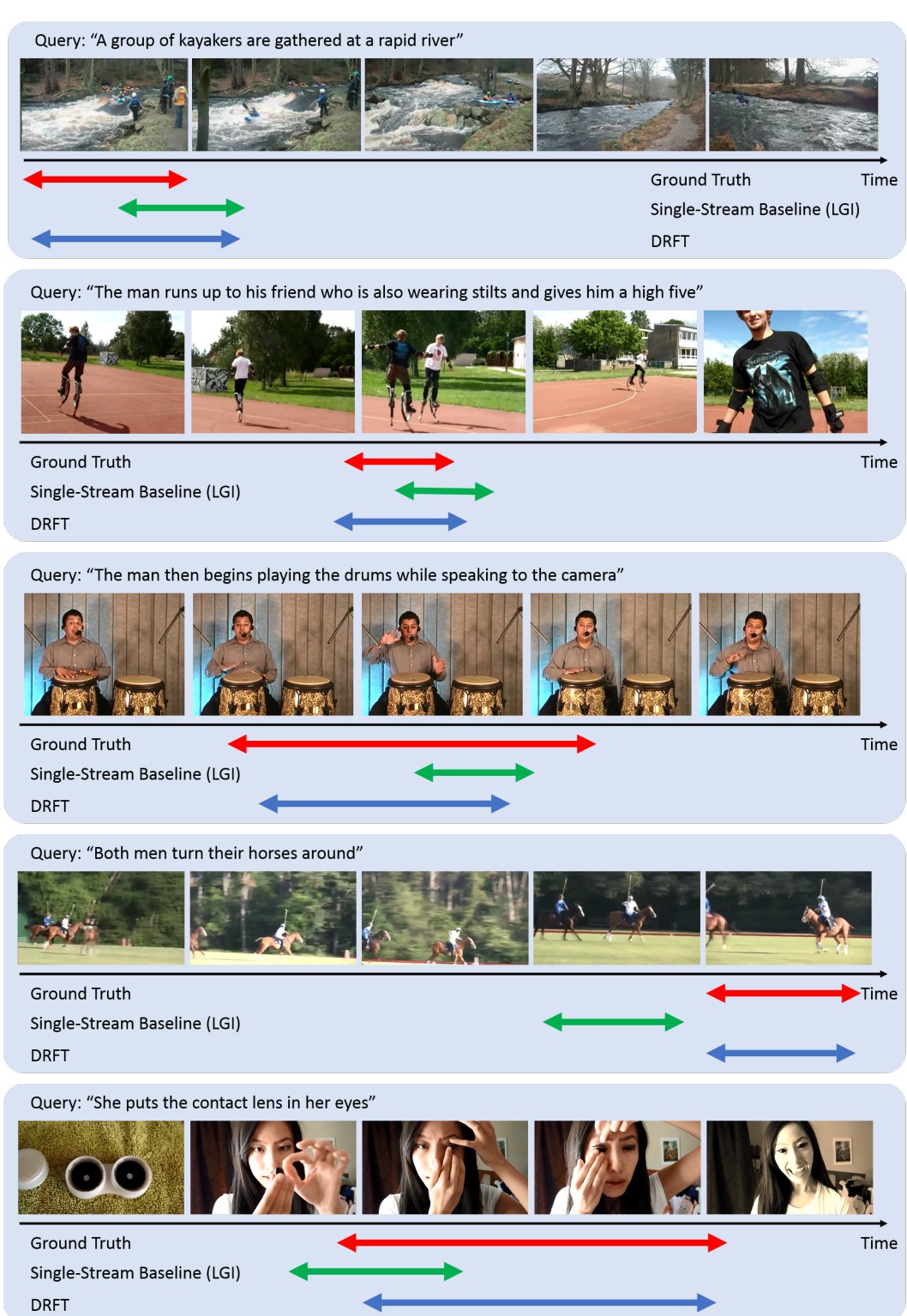

Figure 3: **Sample results on the ActivityNet Captions dataset.** We show that the prediction of the proposed DRFT model is more accurate than the single-stream baseline (LGI [3]) with RGB as input. The arrows indicate the starting and ending points of the grounded video segment based on the query.

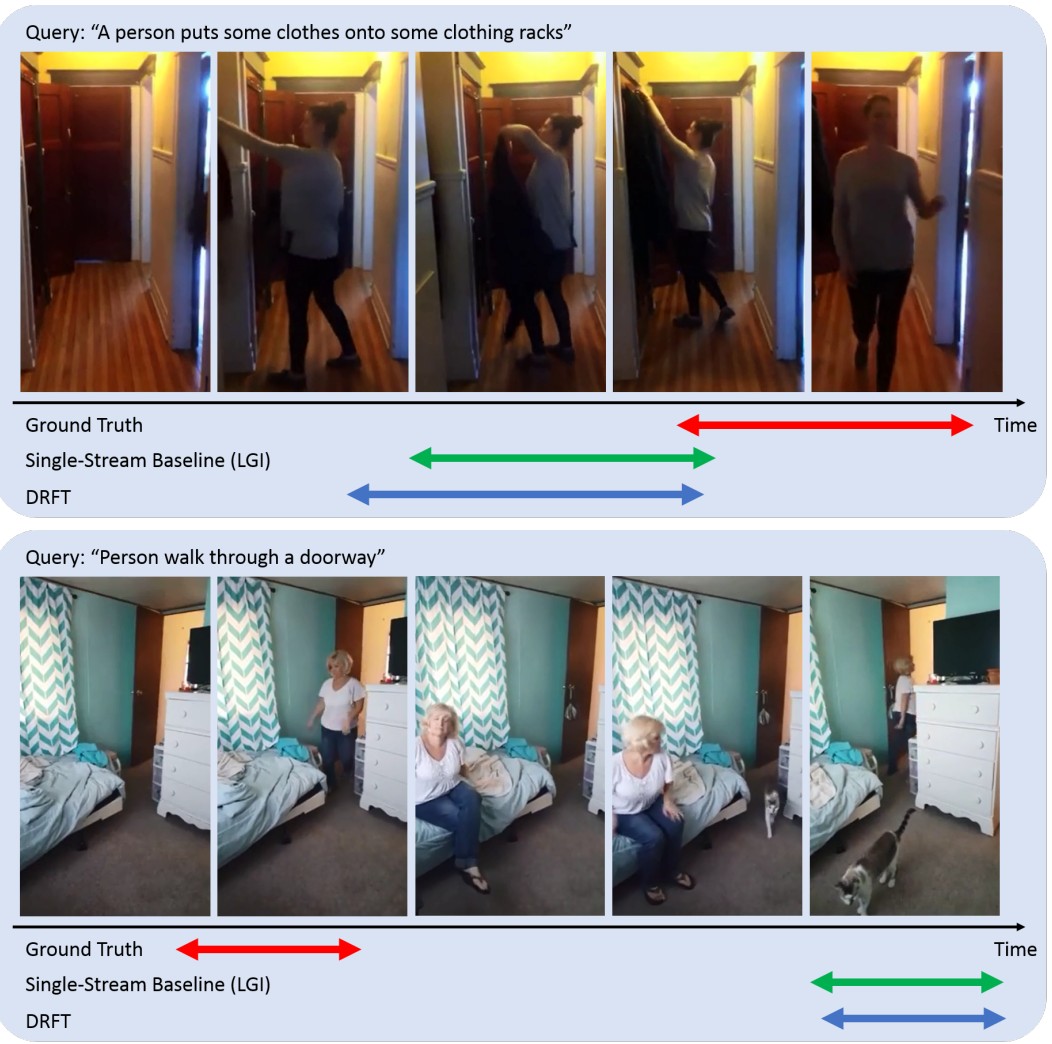

Figure 4: **Failure cases of our algorithm on the Charades-STA dataset.** For the first case, the query event spans longer than the annotation, and our result covers most of the event. In the second video, the event happens twice but only one is annotated, where our model predicts the other one.