# OpenReview forum: "End-to-end Multi-modal Video Temporal Grounding"
_NeurIPS.cc/2021/Conference — NeurIPS 2021 Poster_

### Official Review · Reviewer_wvV3 · 2021-07-11

**Rating:** 4
**Confidence:** 5

**Summary:**

A method for the video temporal grounding in the multi-modal domain (RGB, flow, depth). To this end, the transformer-based co-attention (+ adaptive fusion) and contrastive learning techniques are developed.


**Ethical Concerns:**

no ethical concerns

**Limitations And Societal Impact:**

The authors don't address limitations

**Main Review:**

The proposed modules are straightforward use of existing techniques (transformers, contrastive learning). Although the co-attention scheme is developed by modifying the self-attention scheme of the transformer, it has already been tried in various multi-modal methods. Comprehensively, the novelty is weak. Also, enough analysis and studies are not contained, and several definitions/descriptions (loss term and compared method in analysis) are unclear.

Not enough analysis and studies and several unclear definitions/descriptions, although there is room for the amount of pages.
1) In table 1, does the single-stream DRFT means replacing the inter-modal feature learning module with self-attention? If not, the single-stream DRFT is just the method w/o any attention mechanism, and then it is not proper to verify the effect of the proposed co-attention.

2) Unclear description for REG module. For example, how the start and end timing is obtained if there is multiple start-end in a video?

3) why the definition/formulation of L_grn is not described?

4) in line 205-206, the authors mentioned that their LGI module is similar to LGI model [19]. Then, is the proposed method built on top of [19] by adding the proposed modules?

5) There is no analysis or studies for if the weight of dynamic fusion differs depending on videos or frames. Without showing this point, the reviewer thinks that the gain of dynamic fusion results from merely using more computational cost.

6) Is there no previous multi-modal methods for this task? As the multi-modal baseline, the authors use the simple combination of independently learned uni-modal models. But, the lower performance of the baseline is not surprising at all. To show the effectiveness of the proposed multi-modal fusion, it is required to compare the proposed fusion with other (even basic) intermediate fusion schemes.


**Time Spent Reviewing:**

1 day

---

> ### Author Response · Authors · 2021-08-10
> **Response for Reviewer wvV3**
>
> **Novelty**
>
> While some components of the proposed framework are adopted in previous methods, this work is the first attempt for text-guided video temporal grounding by multi-modal learning. Different from the co-attentional transformer layers [17] and other mechanisms that typically fuse two modalities, we learn complementary information from **three modalities**. To effectively fuse these modalities, we show the following properties in this work:
>
> * We find that the **RGB information is essential** for the temporal grounding task, and thus we conduct co-attention between 1) RGB-flow and 2) RGB-depth. In the table below (row 1), if using the flow modality as the common modality, i.e., flow-RGB and flow-depth, the performance is worse than our final model.
>
> * Since RGB features are used for both flow and depth attention, we adopt a **shared co-attention block** for RGB as shown in Figure 2, where it can take RGB together with either the flow or depth cue as the input, and further enriches the attention mechanism. This design has not been considered in the prior work. In the table below (row 2), without sharing the co-attentional module, the performance is worse than our final model.
>
> * Finally, we show that the proposed **dynamic fusion scheme via learnable weights** is important to fuse features from different modalities. As shown in Table 3 and also the table below (row 3), the performance drops significantly without learnable weights. More insights are provided in another response below.
>
> | | | | | | | | | |
> | :----: | :----: | :----: | :----: | :----: | :----: | :----: | :----: | :----: |
> | | Charades-STA | | | | ActivityNet | Captions | | |
> | Method | R@ 0.3 | R@ 0.5 | R@ 0.7 | mIoU | R@ 0.3 | R@ 0.5 | R@ 0.7 | mIoU |
> | DRFT w/ flow-RGB, flow-depth | 74.96 | 61.20 | 38.57 | 53.01 |  61.39 | 43.94 | 25.85 | 44.17 |
> | DRFT w/o Co-TRM weight sharing | 76.24 | 62.61 | 39.60 |  54.47 | 62.59 | 45.28 | 27.36 | 45.31 |
> | DRFT w/o learnable weights | 75.03 | 61.65 | 38.78 | 53.11 | 61.47 | 44.42 | 26.31 | 44.39 |
> | Three-stream DRFT | 76.68 | 63.03 | 40.15 | 54.89 | 62.91 | 45.72 | 27.79 | 45.86 |
> | | | | | | | | | |
>
> Although intra-modal contrastive learning has been recently used in self-supervised representation learning, it has not been explored in the task of text-guided video temporal grounding to exploit complementary multi-modal features. It requires our careful designs such as **sample construction** and **which features to apply the loss** for effective feature learning:
>
> * We apply contrastive learning on the multi-modal features where textual features attend to visual features of each modality.
>
> * The multi-modal features contain information from the whole video, while we only consider features within a video segment that contains the described action. Thus, an average pooling scheme on the corresponding time interval is used to obtain the features for contrastive learning.
>
> **Single-stream DRFT**
>
> As all the models in Table 1 are baselines without co-attention mechanisms, we think the reviewer means Table 2. Since co-attention is used to fuse different modalities, it cannot be used in the single-stream DRFT. To show the effectiveness of the proposed co-attention, we present in the ablation study of Table 3, which is also shown below. By adding the co-attentional transformer in the full model using three modalities, the performance improves by around 2\%.
>
> | | | | | | | | | |
> | :----: | :----: | :----: | :----: | :----: | :----: | :----: | :----: | :----: |
> |         |  Charades-STA | | | |     ActivityNet | Captions | | |
> | Method  | R@ 0.3 | R@ 0.5 | R@ 0.7 | mIoU | R@ 0.3 | R@ 0.5 | R@ 0.7 | mIoU |
> | DRFT w/o co-attention | 74.72 | 61.05 | 38.26 | 52.74 | 61.04 | 43.83 | 25.74 | 43.90 |
> | Three-stream DRFT | 76.68  |  63.03  |  40.15 |  54.89 |  62.91 |   45.72  |  27.79   | 45.86 |
> | | | | | | | | | |
>
> **REG module**
>
> As described in Ln 206-208, the REG module generates temporal attention weights to aggregate the features and performs regression via an MLP layer to generate the starting and ending time. The operations are defined similar to LGI [19]:
>
> $\textbf{o} = softmax(MLP_{att}(\textbf{R}))$,
>
> $\textbf{v} = \sum \limits_{i=1}^T \textbf{o}_i \textbf{R}_i$,
>
> $t^s, t^e = MLP_{reg}(\textbf{v})$,
>
> where $\textbf{R} \in \mathbb{R}^{d \times T}$ is the output of dynamic feature fusion, $\textbf{o} \in \mathbb{R}^T$ denotes the attention weights for segments, and $\textbf{v} \in \mathbb{R}^d$ indicates the video feature with weighted sum of $\textbf{R}$.
>
> Since this task only considers cases with one starting and ending time given a video and a query sentence, we follow the standard setting and do not consider cases with multiple starting and ending time.
>
> **Supervised loss $L_{grn}$**
>
> The supervised loss is the same as the loss defined in the LGI method [19], which includes:
>
> 1) location regression loss $L_{reg} = smooth_{L1} (\hat{t^s}-t^s) + smooth_{L1} (\hat{t^e}-t^e)$ that calculates the L1 distance between the normalized ground truth time interval $(\hat{t^s}, \hat{t^e}) \in [0, 1]$ and the predicted time interval $(t^s, t^e)$, where $smooth_{L1}$ is defined as $0.5x^2$ if $|x| < 1$ and $|x| - 0.5$ otherwise.
>
> 2) temporal attention guidance loss $L_{tag} = - {\sum^T  \hat{\textbf{o}}_i \log{\textbf{o}_i}} / {\sum^T \hat{\textbf{o}}_i}$ for the temporal attention in the REG module, where $\hat{\textbf{o}}_i$ is set to 1 if the $i$-th segment is located within the ground truth time interval and 0 otherwise.
>
> 3) distinct query attention loss $L_{dqa} = ||(\mathbf{A}^\top \mathbf{A}) - \lambda I||_F^2$ to enforce query attention weights to be distinct along different steps in the LGI module, where $\mathbf{A}\in \mathbb{R}^{L \times N}$ is the concatenated query attention weights across $N$ steps, $||\cdot||_F$ denotes Frobenius norm of a matrix, and $\lambda \in [0, 1]$ controls the extent of overlap between query attention distributions.
>
> The supervised loss is the sum of the three loss terms $L_{grn} = L_{reg} + L_{tag} + L_{dqa}$ and we use the default setting in LGI [19]. We will clarify this part in the paper.
>
> **LGI module**
>
> We use the LGI model [19] as our feature extraction backbone, where textual features attend to visual features, thus obtaining the features for each visual modality. Upon this feature extractor, we design our co-attentional and dynamic feature fusion mechanisms for inter-modal feature learning, as well as the intra-modal contrastive feature learning.
>
> **Weights of dynamic fusion**
>
> Figure 1 of the supplementary material shows the performance of the single-stream DRFT models with RGB, flow or depth as input. The three plots contain action categories where a single modality performs better than the other two modalities.
>
> To further analyze the effect of each modality, we provide the learned weights for dynamic fusion in the three-stream DRFT for these categories in the table below, where the top, middle and bottom groups contain categories that RGB, flow and depth help the most respectively. ‘’Flow $\rightarrow$ RGB’’ means flow-conditioned RGB features, etc. We observe that for actions with smaller movement (top group), the weights for RGB features are larger. For actions with larger motion (middle group), the weights for optical flow are larger. Regarding actions with small motion but can be easily recognized by their structure (bottom group), the weights for depth are larger. This shows that the model can exploit each modality based on the complementary property between RGB, flow and depth.
>
> ||||||
> | :----: | :----: | :----: | :----: | :----: |
> | Action category | Flow $\rightarrow$ RGB | RGB $\rightarrow$ Flow | Depth $\rightarrow$ RGB | RGB $\rightarrow$ Depth |
> | Turning off a light | **0.384** | 0.208 | 0.261 | 0.147 |
> | Someone is smiling | **0.327** | 0.215 | 0.293 | 0.165 |
> | Holding a phone/camera | **0.353** | 0.220 | 0.240 | 0.187 |
> ||||||
> | Closing a door | 0.305 | **0.339** | 0.217 | 0.139 |
> | Throwing a pillow | 0.312 | **0.379** | 0.194 | 0.115 |
> | Throwing a bag somewhere | 0.280 | **0.362** | 0.203 | 0.155 |
> ||||||
> | Sitting in a bed | 0.205 | 0.185 | 0.236 | **0.374** |
> | Sitting in a chair | 0.221 | 0.173 | 0.284 | **0.322** |
> | Working at a table | 0.216 | 0.158 | 0.262 | **0.364** |
> ||||||
>
> **Comparison with other fusion schemes**
>
> To the best of our knowledge, this work is the first attempt for text-guided video temporal grounding by multi-modal learning. In the baseline models, we take average of features from different modalities since it is also a common setting in multi-modal learning methods for other tasks such as [A, B]. We show results of another baseline in the table below with slightly decreased performance, where features of different modalities are concatenated and then fused by a $1 \times 1$ convolution. We hypothesize that, since we have more than two modal features to fuse, using additional layers for fusion needs further design considerations. In contrast, with the proposed co-attentional and dynamic fusion mechanisms in the full model, we can further improve the performance.
>
> [A] Munro et al. “Multi-Modal Domain Adaptation for Fine-Grained Action Recognition.” In CVPR 2020.
>
> [B] Simonyan and Zisserman. "Two-Stream Convolutional Networks for Action Recognition in Videos." In NeurIPS 2014.
>
> | | | | | | | | | |
> | :----: | :----: | :----: | :----: | :----: | :----: | :----: | :----: | :----: |
> | | Charades-STA | | | | ActivityNet | Captions | | |
> | Method  | R@ 0.3 | R@ 0.5 | R@ 0.7 | mIoU | R@ 0.3 | R@ 0.5 | R@ 0.7 | mIoU |
> | Concat w/ $1 \times 1$ conv | 70.86 | 56.91 | 34.30 | 47.78 | 56.18 | 38.12 | 21.66 | 39.45 |
> | Three-stream baseline | 71.13 | 57.39 | 34.69 | 48.21 | 56.45 | 38.63 | 22.05 | 39.86 |
> | Three-stream DRFT | 76.68  |  63.03  |  40.15 |  54.89 |  62.91 | 45.72 |  27.79 | 45.86 |
> | | | | | | | | | |

---

> > ### Author Response · Authors · 2021-08-31
> > **Please let us know whether you have additional questions**
> >
> > Dear Reviewer,
> >
> > We have provided more results and explanations based on your review. Please go over them and let us know whether you have additional questions or not.
> >
> > Thank you,

---

> ### Author Response · Authors · 2021-09-02
> **Please let us know whether you have additional questions**
>
> Dear Reviewer,
>
> We thank your comments and add more results/explanations based on your review. Please go over our response and let us know whether you have additional questions or not.
>
> Thank you,

---

### Official Review · Reviewer_bBpz · 2021-07-15

**Rating:** 6
**Confidence:** 3

**Summary:**

This paper addresses text-guided video grounding, which identifies the time interval of an event according to the query text. In contrast to prior work that only leverages RGB information, the authors propose DRFT to combine RGB, depth, and flow maps to boost the performance. To facilitate representation learning, the authors adopt a co-attentional transformer to fuse multi-modal information and contrastive learning to enhance the features across different videos. Experimental results shows that the proposed DRFT outperforms state-of-the-art on Charades-STA and ActivityNet.

**Limitations And Societal Impact:**

The authors are encouraged to discuss potential abuse and misuse.

**Main Review:**

Overall, the paper is well-written and easy to follow. Experimental results also justify the effectiveness of the proposed method. However, the authors are expected to justify their design choices in the proposed method rather than just give a combination of existing techniques. It would help further extension if the authors could elaborate more on the current design. More details can be found below.

1.	In DRFT, the authors only consider two kinds of fusion, namely RGB-depth and RGB-flow. Is there any specific reason not to use other combinations, e.g., depth-flow?
2.	The baseline models in Sec. 3.1 do not make sense to me. It seems to me that the most straightforward would be to concatenate different modalities and learn a 1x1 conv to fuse those features. Is there any reason that the authors adopt an average strategy? What’s the gap between the concatenation and other models mentioned in the paper?
3.	I am curious about the necessity of using a co-attentional transformer. There have been a few lightweight attention modules, such as CBAM [1]. Can the authors briefly elaborate on this design choice?
4.	Although the experiments suggest multi-modality does help, it is unclear how they help. Maybe the authors could provide visualization showing what kind of information helps which activities more, by showing the attention weights and the corresponding activities.
5.	The reason why intra-model feature learning works remains unclear to me. It might be a novel point of view, but the authors are encouraged to provide more insights in the paper.

[1] Woo, Sanghyun, et al. "Cbam: Convolutional block attention module." Proceedings of the European conference on computer vision (ECCV). 2018.


**Time Spent Reviewing:**

8

---

> ### Author Response · Authors · 2021-08-10
> **Response for Reviewer bBpz**
>
> **Other combinations of fusion**
>
> We include RGB-depth and RGB-flow in the two-stream models because RGB information is important for our task. Our single-stream model using RGB features also performs better than using only optical flow or depth features. In the table below, we provide the results of two-stream models that take depth-flow as input. The performance gap between the depth-flow models and other fusions shows that RGB information is essential in this task. Combining all the modalities with the proposed modules further shows the complimentary property.
>
> | | | | | | | | | | |
> | :----: | :----: | :----: | :----: | :----: | :----: | :----: | :----: | :----: | :----: |
> |     |  |  Charades-STA | | | |     ActivityNet | Captions | | |
> | Method | Visual Feat. | R@ 0.3 | R@ 0.5 | R@ 0.7 | mIoU  | R@ 0.3 | R@ 0.5 | R@ 0.7 | mIoU |
> | Two-stream baseline | R, F | 71.57 | 58.35 | 35.72 | 50.34 | 57.78 | 40.51 | 22.66 | 40.70 |
> | Two-stream DRFT | R, F | 74.26 | 61.93 | 38.69 | 53.92 | 61.80 | 43.71 | 26.43 | 44.82 |
> | | | | | | | | | | |
> | Two-stream baseline | D, R | 71.34 | 57.51 | 35.49 | 48.70 | 57.43 | 40.46 | 22.35 | 39.69 |
> | Two-stream DRFT | D, R | 73.72 | 60.93 | 38.12 | 52.75 | 61.18 | 43.29 | 25.86 | 43.61 |
> | | | | | | | | | | |
> | Two-stream baseline | D, F | 70.94 | 57.21 | 35.08 | 48.56 | 57.13 | 39.82 | 21.79 | 39.56 |
> | Two-stream DRFT | D, F | 72.83 | 59.32 | 37.26 | 50.81 | 59.34 | 42.08 | 23.96 |  41.84 |
> | | | | | | | | | | |
>
>
>
> **Design of baseline models**
>
> We take the averaged output of different modalities in the baseline models as it is also a common setting in other multi-modal learning methods such as [A, B]. As suggested by the reviewer, we conduct the experiment of concatenating different modalities and learning a $1 \times 1$ convolution to fuse the features. The results are provided in the table below. We show that our baseline model with the average strategy slightly performs better than the concatenation method with $1 \times 1$ convolution. Since we have more than two modal features to fuse, using additional layers for fusion needs further design considerations. With the proposed co-attentional and dynamic fusion mechanisms in the full model, we can further improve the performance.
>
> [A] Munro et al. “Multi-Modal Domain Adaptation for Fine-Grained Action Recognition.” In CVPR 2020.
>
> [B] Simonyan and Zisserman. "Two-Stream Convolutional Networks for Action Recognition in Videos." In NeurIPS 2014.
>
> | | | | | | | | | |
> | :----: | :----: | :----: | :----: | :----: | :----: | :----: | :----: | :----: |
> |         |  Charades-STA | | | |     ActivityNet | Captions | | |
> | Method  | R@ 0.3 | R@ 0.5 | R@ 0.7 | mIoU | R@ 0.3 | R@ 0.5 | R@ 0.7 | mIoU |
> | Concat w/ $1 \times 1$ conv | 70.86 | 56.91 | 34.30 | 47.78 | 56.18 | 38.12 | 21.66 | 39.45 |
> | Three-stream baseline | 71.13 | 57.39 | 34.69 | 48.21 | 56.45 | 38.63 | 22.05 | 39.86 |
> | Three-stream DRFT | 76.68  |  63.03  |  40.15 |  54.89 |  62.91 |   45.72  |  27.79   | 45.86 |
> | | | | | | | | | |
>
>
> **Design choice of co-attentional transformer**
>
> We use an approach similar to the co-attentional transformer as it has shown success in several multi-modal learning tasks. Different from the co-attentional transformer in [17] that only considers two modalities, we deal with **three modalities**, RGB, flow and depth. To this end, we also adopt a **shared co-attention block for RGB** as shown in Figure 2 of the main paper, where the co-attentional module for RGB can take either the flow or depth cue, which further enriches the attention mechanism.
>
> CBAM focuses on channel attention and spatial attention in one single modality, while we aim to perform multi-modal learning. Therefore, we do not take CBAM into account in our current setting. We will explore different kinds of attention modules in the future.
>
>
> **How multi-modal learning helps**
>
> The inter-modal feature learning includes 1) **co-attentional feature fusion** across modalities and 2) **dynamic feature fusion** for combining multi-modal features.
>
> * The co-attentional feature fusion is applied on the temporal features with dimension $c \times T$, where $c$ is the feature dimension and $T$ is the number of frames in the whole video. Then we generate the temporal attention weights and try to compare them between the baseline model and our proposed DRFT. We find that the attention weights focus more on the target interval in the proposed DRFT. While we are not able to show the visualization in the rebuttal format, we will add the analysis in the revised manuscript.
>
> * Regarding the dynamic multi-modal feature fusion, in Figure 1 of the supplementary material, we show the performance of the single-stream DRFT models with RGB, flow or depth as input. The three plots contain action categories where a single modality (RGB, flow or depth) performs better than the other two modalities. To further analyze the impact of each modality, we provide the learned weights for dynamic fusion in the three-stream DRFT for these categories in the table below, where the top, middle and bottom groups contain categories that RGB, flow and depth help the most respectively. ‘’Flow $\rightarrow$ RGB’’ means flow-conditioned RGB features, etc. We observe that for actions with smaller movement (top group), the weights for RGB features are larger. For actions with larger motion (middle group), the weights for optical flow are larger. Regarding actions with small motion but can be easily recognized by their structure (bottom group), the weights for depth are larger. This shows that the model can exploit each modality based on the complementary property between RGB, flow and depth.
>
> ||||||
> | :----: | :----: | :----: | :----: | :----: |
> | Action category | Flow $\rightarrow$ RGB | RGB $\rightarrow$ Flow | Depth $\rightarrow$ RGB | RGB $\rightarrow$ Depth |
> | Turning off a light | **0.384** | 0.208 | 0.261 | 0.147 |
> | Someone is smiling | **0.327** | 0.215 | 0.293 | 0.165 |
> | Holding a phone/camera | **0.353** | 0.220 | 0.240 | 0.187 |
> ||||||
> | Closing a door | 0.305 | **0.339** | 0.217 | 0.139 |
> | Throwing a pillow | 0.312 | **0.379** | 0.194 | 0.115 |
> | Throwing a bag somewhere | 0.280 | **0.362** | 0.203 | 0.155 |
> ||||||
> | Sitting in a bed | 0.205 | 0.185 | 0.236 | **0.374** |
> | Sitting in a chair | 0.221 | 0.173 | 0.284 | **0.322** |
> | Working at a table | 0.216 | 0.158 | 0.262 | **0.364** |
> ||||||
>
>
>
> **​​Intra-modal feature learning**
>
> Given one modality, the motivation of our intra-modal feature learning is that features in the same action category should be similar even if they are from different videos.
>
> We also note that our proposed intra-modal contrastive learning has not been explored in the task of text-guided video temporal grounding to exploit complementary multi-modal features. It requires our careful designs such as **sample construction** and **which features to apply the loss** for effective feature learning:
>
> * We apply contrastive learning on the multi-modal features where textual features attend to visual features of each modality.
>
> * The multi-modal features contain information from the whole video, while we only consider features within a video segment that contains the described action. Therefore, an average pooling scheme on the corresponding time interval is used to obtain the features for contrastive learning.

---

### Official Review · Reviewer_wDga · 2021-07-16

**Rating:** 7
**Confidence:** 4

**Summary:**

Paper addresses the problem of text-guided video temporal grounding, which aims to localize the starting and ending time of a segment corresponding to a text query. The key contribution is integration of three modalities of data in this context: video, motion (flow), and depth; along with the textural query. To integrate the three modalities, paper proposes a dynamic fusion scheme with transformers, which takes the form of co-attention (similar to ViLBERT). Further, to improve the performance within each modality contrastive learning is applied to enhance the feature discriminability. In this contrastive learning formulation positive pairs come from instance of the same action class, while negative pairings are formed from videos coming from two different action classes. Competitive (state-of-the-art) performance is illustrated on Charades-STA and ActivityNet Captions benchmark datasets.

**Ethical Concerns:**

None.

**Limitations And Societal Impact:**

Yes. I do not believe there are deep societal impacts to discuss for this work.

**Main Review:**

The paper is well written and the approach is intuitive and easy to understand and follow. Performance is also competitive and improves on state-of-the-art.  The technical novelty is somewhat increment, with components effectively borrowed from other recent works. That being said the choices are well motivated and work well together for the task at hand. Overall, I feel the novelty is sufficient for a poster publication in NeurIPS.

A few minor comments

- The supervised loss (L_{and}) is not defined. It should be defined in the paper, if for no other reason than completeness.
- The extraction of video segment features (Lines 188-190) is unclear and should be clarified.
- In addition, one of the more surprising aspects of the paper, for me, is that depth was helpful for the task. I am not aware of any other works that use estimated depth for video tasks. As such, I would be interested in seeing more analysis on the depth modality. For example, how well would depth features work on their own (as one stream DRFT)? How well would they perform in combination with RGB (as a two stream DRFT)? etc.

**Time Spent Reviewing:**

6

---

> ### Author Response · Authors · 2021-08-10
> **Response for Reviewer wDga**
>
> **Supervised loss $L_{grn}$**
>
> The supervised loss is the same as the loss defined in the LGI method [19], which includes:
>
> 1) location regression loss $L_{reg} = smooth_{L1} (\hat{t^s}-t^s) + smooth_{L1} (\hat{t^e}-t^e)$ that calculates the L1 distance between the normalized ground truth time interval $(\hat{t^s}, \hat{t^e}) \in [0, 1]$ and the predicted time interval $(t^s, t^e)$, where $smooth_{L1}$ is defined as $0.5x^2$ if $|x| < 1$ and $|x| - 0.5$ otherwise.
>
> 2) temporal attention guidance loss $L_{tag} = - {\sum^T  \hat{\textbf{o}}_i \log{\textbf{o}_i}} / {\sum^T \hat{\textbf{o}}_i}$ for the temporal attention in the REG module, where $\hat{\textbf{o}}_i$ is set to 1 if the $i$-th segment is located within the ground truth time interval and 0 otherwise.
>
> 3) distinct query attention loss $L_{dqa} = ||(\mathbf{A}^\top \mathbf{A}) - \lambda I||_F^2$ to enforce query attention weights to be distinct along different steps in the LGI module, where $\mathbf{A}\in \mathbb{R}^{L \times N}$ is the concatenated query attention weights across $N$ steps, $||\cdot||_F$ denotes Frobenius norm of a matrix, and $\lambda \in [0, 1]$ controls the extent of overlap between query attention distributions.
>
> The supervised loss is the sum of the three loss terms $L_{grn} = L_{reg} + L_{tag} + L_{dqa}$ and we use the default setting in LGI [19].
>
>
> **Extraction of video segment features**
>
> We consider the multi-modal features $M_d$, $M_r$ and $M_f$, each with dimension $c \times T$ for the intra-modal feature learning, where $c$ is the feature dimension and $T$ is the number of frames in the whole video. Since the action is only contained in a segment of the video, we use the starting and ending time $(\hat{t^s}, \hat{t^e})$ of the action to obtain the corresponding features with dimension $c \times t$, where $t = \hat{t^e} - \hat{t^s}$. We then conduct average pooling in the temporal dimension and obtain the features with dimension $c$ for contrastive loss.
>
>
> **More analysis on the depth modality**
>
> We present the results of single-stream DRFT with depth features and two-stream DRFT with depth-RGB or depth-flow in the table below. Results of the first two settings are also provided in the supplementary material. We observe that, compared to the other two modalities, **depth is not the strongest one but is still complementary to other modalities**. Moreover, the proposed DRFT method can always improve the baseline performance. In Figure 1 of the supplementary material, we show some sample action categories where depth information helps more than RGB and flow. These actions are usually with small motion, but involve objects that can be easily recognized by their structure, e.g., “*sitting in a bed*”, “*sitting in a chair*” or “*working at a table*”.
>
>
> | | | | | | | | | | |
> | :----: | :----: | :----: | :----: | :----: | :----: | :----: | :----: | :----: | :----: |
> |     |  |  Charades-STA | | | |     ActivityNet | Captions | | |
> | Method | Visual Feat. | R@ 0.3 | R@ 0.5 | R@ 0.7 | mIoU  | R@ 0.3 | R@ 0.5 | R@ 0.7 | mIoU |
> | Single-stream baseline | D | 70.17 | 55.93 | 34.20 | 47.68 | 56.52 | 39.23 | 21.26 | 38.47 |
> | Single-stream DRFT | D | 71.54 | 57.78 | 35.92 | 49.55 | 57.59 | 40.64 | 22.61 | 40.12 |
> | Two-stream baseline | D, R | 71.34 | 57.51 | 35.49 | 48.70 | 57.43 | 40.46 | 22.35 | 39.69 |
> | Two-stream DRFT | D, R | 73.72 | 60.93 | 38.12 | 52.75 | 61.18 | 43.29 | 25.86 | 43.61 |
> | Two-stream baseline | D, F | 70.94 | 57.21 | 35.08 | 48.56 | 57.13 | 39.82 | 21.79 | 39.56 |
> | Two-stream DRFT | D, F | 72.83 | 59.32 | 37.26 | 50.81 | 59.34 | 42.08 | 23.96 |  41.84 |
> | | | | | | | | | | |

---

### Official Review · Reviewer_9jTB · 2021-07-17

**Rating:** 6
**Confidence:** 4

**Summary:**

This paper studies the problem of multi-modal video grounding. The authors exploit multiple modalities including RGB, depth, optical flow to extract complementary information for improving video grouding. A dynamic fusion scheme with transformer is proposed to better learn the interaction and integration of multiple modalities. Moreover, a self-supervised intrao-modal module is proposed to obtain better feature representation. The extensive experiments support that the proposed method achieves new state-of-the-art performance on multiple bechmark datasets.

**Limitations And Societal Impact:**

No limitations provided. No more suggestions.

**Main Review:**

Strengths:
1. Satisfactory paper writing. Clear technical presentation.
2. Good performances and thorough empirical studies.

Weaknesses:
1. My main concern is the novelty. All the proposed modules look natural but kind of straighforward to me. They indeed contribute to the performance, but there is no inspiring techniques or insightful technical conclusion. For example, co-attentional transformer layers are adopted from [17] to model the interaction among different modalities; contrastive learning is adopted to model the intra-modal feature learning.

Questions:
1. Why actions like "sitting in a bed" require depth information? Can they be recognized by RGB information?
2. In the intra-modal feature learning, given only the sentence query, how to judge whether two videos are from the same action category?

Rebuttal summary:
1. I agree with other reviewers that depth information is interesting and properly leveraged for the multi-modality task. From non-technical aspect, it is a novel idea.
2. The technical novelty might be limited but acceptable. I increase my rating to "Marginally above acceptence threshold".

**Time Spent Reviewing:**

2

---

> ### Author Response · Authors · 2021-08-10
> **Response for Reviewer 9jTB**
>
> **Novelty**
>
> While some components of the proposed framework are adopted from previous methods, this work is the first attempt for text-guided video temporal grounding by multi-modal learning. Different from the co-attentional transformer layers in [17] and other mechanisms that typically fuse two modalities, we learn complementary information from **three modalities**, including RGB, optical flow and depth. To effectively fuse these modalities, we show the following properties in this work:
>
> * We find that the **RGB information is essential** for the temporal grounding task, and thus we conduct co-attention between 1) RGB-flow and 2) RGB-depth. In the table below (row 1), if using the flow modality as the common modality, i.e., flow-RGB and flow-depth, the performance is worse than our final model.
>
> * Since RGB features are used for both flow and depth attention, we adopt a **shared co-attention block** for RGB as shown in Figure 2 of the main paper, where it can take RGB together with either the flow or depth cue as the input, which further enriches the attention mechanism. This design has not been considered in the prior work. In the table below (row 2), we show that without sharing the co-attentional module, the performance is worse than our final model.
>
> * Finally, we show that the proposed **dynamic fusion scheme via learnable weights** is important to fuse features from different modalities. As shown in Table 3 of the main paper and also the table below (row 3), the performance drops significantly without learnable weights. More insights are provided in another response below.
>
> | | | | | | | | | |
> | :----: | :----: | :----: | :----: | :----: | :----: | :----: | :----: | :----: |
> |         |  Charades-STA | | | |     ActivityNet | Captions | | |
> | Method  | R@ 0.3 | R@ 0.5 | R@ 0.7 | mIoU | R@ 0.3 | R@ 0.5 | R@ 0.7 | mIoU |
> DRFT w/ flow-RGB, flow-depth | 74.96  |  61.20  |  38.57 |  53.01 |  61.39  |  43.94  |  25.85  | 44.17 |
> DRFT w/o Co-TRM weight sharing  | 76.24  |  62.61  |  39.60 |  54.47 |  62.59 |   45.28  |  27.36 |  45.31 |
> DRFT w/o learnable weights | 75.03  |  61.65  |  38.78 |  53.11 |  61.47  |  44.42  |  26.31 |  44.39 |
> Three-stream DRFT | 76.68  |  63.03  |  40.15 |  54.89 |  62.91 |   45.72  |  27.79   | 45.86 |
> | | | | | | | | | |
>
>
> For the proposed intra-modal contrastive learning, although it is often used in self-supervised representation learning, it has not been explored in the task of text-guided video temporal grounding to exploit complementary multi-modal features. It requires our careful designs such as **sample construction** and **which features to apply the loss** for effective feature learning:
>
> * We apply contrastive learning on the multi-modal features where textual features attend to visual features of each modality.
>
> * The multi-modal features contain information from the whole video, while we only consider features within a video segment that contains the described action. Therefore, an average pooling scheme on the corresponding time interval is used to obtain the features for contrastive learning.
>
>
> **​​Depth vs. RGB information**
>
> While RGB features provide abundant visual cues, they may be affected by background clutters. In contrast, depth information is invariant to color and lighting. Therefore, it can help more in actions such as ‘’*sitting in a bed*’’ which involve objects that can be easily distinguished by their shapes. In the table below, we show the learned weights in the dynamic fusion module for the action ‘’*sitting in a bed*’’. The ‘’$\rightarrow$’’ means attention from one to another modality. The RGB-conditioned depth features (RGB $\rightarrow$ Depth) have the largest weight, which indicates that depth information is the most important cue in this action. In the third plot of Figure 1 in the supplementary material, we also show that the depth-only model performs better than the RGB-only model in this action.
>
> ||||||
> | :----: | :----: | :----: | :----: | :----: |
> | Action category | Flow $\rightarrow$ RGB | RGB $\rightarrow$ Flow | Depth $\rightarrow$ RGB | RGB $\rightarrow$ Depth |
> | Sitting in a bed | 0.205 | 0.185 | 0.236 | **0.374** |
> ||||||
>
>
> **Action category in the intra-modal feature learning**
>
> We calculate the similarity between the features of query sentences to determine the action categories. Since many of the query sentences contain the action labels as indicated in the ActivityNet Captions paper [14], the textual features of the same action category are usually close to each other in the feature space and therefore the action category can be easily determined.

---

### Decision · Program_Chairs · 2021-09-27

**Decision:**

Accept (Poster)

**Comment:**

This paper introduces an approach for multi-modal video temporal grounding. The paper received diverging review scores (4, 6, 6, 7),
with the positive reviewer championing the paper. All reviewers agree that the approach is somewhat incremental, but it introduces as one of the first the use of depth information. The rebuttal addressed a significant part of the concerns of the reviewers.

The recommendation is accept as a poster, with the expectation that the authors include the information provided in the rebuttal.